# Selection and Fitting of Mixed Models in Sugarcane Yield Trials

**Josafhat Salinas-Ruíz [1], Sandra Luz Hernández-Valladolid [2], Juan Valente Hidalgo-Contreras [1]** 
**and Juan Manuel Romero-Padilla [3],***

1   Colegio de Postgraduados, Campus Córdoba, Carretera Córdoba-Veracruz Km. 348, Manuel León,
    Amatlán de los Reyes 94953, Veracruz, Mexico; salinas@colpos.mx (J.S.-R.); jvhidalgo@colpos.mx (J.V.H.-C.)
2   Agricultura Sustentable y Protegida, Universidad Tecnológica del Centro de Veracruz,
    Avenida Universidad 350, Cuitláhuac 94910, Veracruz, Mexico; sandra.hernandez@utcv.edu.mx
3   Colegio de Postgraduados, Campus Montecillo, Carretera México-Texcoco Km. 36.5, Montecillo,
    Texcoco 56230, Estado de México, Mexico
*   Correspondence: romero.manuel@colpos.mx; Tel.: +52-59-5952-0200

**Abstract:** Mixed models are a useful tool for the analysis of sugarcane field trials in which sugarcane varieties are allocated in different locations and phenotypic traits are evaluated in the same experimental unit (plot) over time. One challenge to analyze these data is how to build a good mixed model when no experimental design is planned, because all sugarcane varieties in the area of influence of a sugar mill are planted in different years due to the age of the crop and there is no spatial information on all plots. The aim of this research was to examine and to determine the most appropriate mixed model for estimating cane stalk yield of sugarcane varieties when previously there was no planned experimental design. Cane stalk yields of 26 sugarcane genotypes harvested in 24 different locations and in different crop cycles (age) were analyzed. The randomized block nested design (plot within block) with ratoon crop as a class variable in the mixed model was the best for the mean comparisons in sugarcane genotype trials (Model 3), allowing a gain in information. The randomized block design approach helps to fit more general random effects, and the covariance structures helps to improve the performance of mixed model repeated measures. This study emphasizes the need to improve the process of finding a good enough mixed model, that is, how to define the mean structure and the best covariance structure for model sugarcane trials that enables more powerful and efficient parameter estimations. The results showed how a more appropriate mixed model might help avoid errors of judgment in sugarcane genotype recommendations for enhancing the productivity of the cane industry.

**Keywords:** mixed model; fit statistics; covariance structure; mean square error

## 1. Introduction

Sugarcane is a very important crop cultivated in tropic regions. In the crop year 2018–2019 [1], sugarcane was cultivated on 805,500 ha in 22 states of Mexico; in particular, it was cultivated on 302,062 ha in the state of Veracruz (37.5% of total area). Due to the social and economic impact of this crop for Veracruz, the government and sugar agro-industries need good estimates, such as stalk cane yield of sugarcane varieties, to make important economic decisions that benefit the local agro-industry.

A large number of sugarcane varieties are planted and harvested every year from fields and then sent to a nearby sugar mill. Yields of these genotypes are affected by several factors such as variety, location, and number of ratoons. Ratooning refers to harvesting for several years from the same plant, and it plays an important role in sugarcane production. Sugarcane crops are harvested sequentially from the initial plant, first ratoon, second ratoon, and so on, in successive years. High-yield cane, high ratooning ability, and wide adaptability are important factors for the assessment of good profitability in sugarcane production [2,3]. Planting a new sugarcane crop is expensive, and it requires transporting

large quantities of vegetative planting material from the source field to the new fields. Thus, maintaining ratoon crops is better than replacing all plants or planting a new variety each year. Moreover, land preparation, planting, and irrigation systems, if available, increase the cost of establishing this crop. Hence, ratooning ability in a sugarcane crop is an important factor that leads to harvesting more or less stalks of cane per plant.

Data on multiple traits measured over time are correlated; for example, if ratooning is used, the cane crop yields in year 1 and year 2 are highly correlated [4]. When this correlation is ignored, the experimental error may be underestimated or overestimated. Thus, the analysis of yield data needs to consider the plot correlation and the correlation of the variables measured across crop-years. Repeated measures analysis may consider multiple correlations. Repeated measures experiments are very common in the agricultural and animal sciences [4,5]. The usual way is to apply treatments to experimental units in a completely randomized design or randomized complete block design, and measurements are performed sequentially over time. When several measurements are taken from the same experimental unit, pairs of measurements tend to be highly correlated between themselves. Such measurements are taken on plots that have been randomly allocated treatments as fixed effects such as crop variety with different locations and replications as random effects [5].

Applications of the mixed model methodology have been carried out in sugarcane trials to model the genotype–environment interaction to study the behavior of the varieties and their suitability in each environment. For the selection of sugarcane genotypes for local adaptations with the aim of improving the average productivity and profitability of the entire cane industry, the linear mixed models (LMMs) play an important role. Menezes et al. [6] used the harmonic mean through the mixed model framework for selecting clones with the best genotypic stability and adaptability for each location, and Ostengo et al. [7] showed that use of an appropriate mixed model to analyze tons of cane per hectare and sucrose content when there is a spatial variation would help avoid judgment errors in sugarcane recommendations. Thus, there is great practical interest in simultaneously identifying the important predictors that correspond to both the fixed and random effects components in a linear mixed-effects model. Most approaches often perform selection separately on each of the fixed and random effect components. However, changing the structure of the systematic and random structure in the model can lead to different results. Therefore, sugarcane variety, locations, ratooning, time, and, even more, the lack of experimental design, are key factors to be taken into account for estimating and predicting sugarcane yield. The aim of this work was to evaluate different linear mixed models that enable the analysis of cane yield and to show how the mean structure in the model may give different results.

## 2. Materials and Methods

### 2.1. Plant Material and Data Collection

An on-farm experiment was conducted. An agreement was drawn up with farmers to allow data collection from their farming plots; the experimental units were crop plots of different sizes. A sugar mill received all mill-able cane stalks, which were burned in order to remove dry leaves and weighed per plot (Mg ha$^{-1}$). The numbers of units (plots) for each location are shown in Table 1.

Crop cycles 2012–2013, 2013–2014, 2014–2015, and 2015–2016 were evaluated, and one harvest per crop cycle was obtained; because in some plot units, information was not collected during the four evaluation periods, data were unbalanced. The sugarcane stalk yields from 26 varieties (ATEMEX 96-40, B 43-62, BCO. DE VARS, CP 44-101, CP 70-1133, CP 72-1210, CP 72-2086, CO 997, ITV 92-1424, ITV 92-373, L 77-50, LGMEX 92-156, M.Y. 5514, MEX 56-18, MEX 56-476 (P-01), MEX 57-473, MEX 68-P-23, MEX 69-290, MEX 73-523, MEX 79-431, MEZCLA PREC, MEZCLA MEDIA, P.O.J 2878, PR 1013, RD 75-11, SP 70-1284) were harvested in 24 different zones (locations) from the central region of the state of Veracruz: 18°53′05″ N and 96°47′15″ W, and 503 masl. These locations represent the

whole area in which sugar cane varieties were planted with different numbers of replicates, different plot sizes and different ratoon crops in each production cycle. The information on 16,836 plot units in 812 large fields in 24 locations was collected in four crop cycles with a total of 44,589 records. Table 2 shows the plots planted using plant-cane at the beginning of each production cycle and the number of plots with ratoon number ($R_i$) at harvest.

**Table 1.** Number of fields and plots by location and average surface plots.

| Location Code | Fields | Plots | Average Surface of Plots (ha) |
|:---:|:---:|:---:|:---:|
| 11 | 59 | 840 | 1.91 |
| 12 | 51 | 726 | 1.60 |
| 13 | 74 | 1379 | 1.69 |
| 14 | 47 | 992 | 1.46 |
| 21 | 34 | 925 | 1.38 |
| 22 | 27 | 1145 | 1.29 |
| 23 | 36 | 828 | 1.53 |
| 24 | 26 | 275 | 1.87 |
| 31 | 31 | 824 | 1.52 |
| 32 | 35 | 798 | 1.99 |
| 33 | 41 | 1139 | 1.49 |
| 34 | 31 | 769 | 1.92 |
| 35 | 27 | 893 | 1.43 |
| 36 | 13 | 238 | 4.35 |
| 37 | 11 | 49 | 5.88 |
| 41 | 37 | 966 | 1.89 |
| 42 | 33 | 644 | 2.42 |
| 43 | 28 | 745 | 2.31 |
| 44 | 30 | 653 | 2.36 |
| 45 | 24 | 553 | 2.51 |
| 46 | 23 | 279 | 2.28 |
| 47 | 34 | 342 | 2.86 |
| 48 | 36 | 485 | 2.20 |
| 51 | 24 | 349 | 2.39 |

**Table 2.** Number of plots having the $R_i$ ratoon in each production cycle.

| Production Cycle | Ratoon Number | | | | | | | | |
|:---:|:---:|:---:|:---:|:---:|:---:|:---:|:---:|:---:|:---:|
| | $R_0$ | $R_1$ | $R_2$ | $R_3$ | $R_4$ | $R_5$ | $R_6$ | $R_7$ | $R_8$ |
| 2012–2013 | 555 | 476 | 456 | 1330 | 259 | 6352 | 0 | 0 | 0 |
| 2013–2014 | 1055 | 473 | 507 | 612 | 234 | 295 | 6301 | 0 | 0 |
| 2014–2015 | 1401 | 1885 | 731 | 600 | 309 | 339 | 425 | 7105 | 0 |
| 2015–2016 | 782 | 1048 | 1805 | 724 | 585 | 334 | 404 | 439 | 6767 |

### 2.2. Statistical Methodology

*Linear Mixed Model.* The mixed model methodology has advantages over fixed linear models [4] because it can incorporate fixed effects and random effects; the mixed model effects are given by:

$$Y = X\beta + Zb + \varepsilon \tag{1}$$

with $b \sim N(\mathbf{0}, D)$ and $\varepsilon \sim N(\mathbf{0}, R)$.

where $Y$: vector of observations; $X$: matrix of fixed effects; $\beta$: vector of coefficients of fixed effects; $Z$: matrix of random effects; $b$: vector of coefficients of random effects; $\varepsilon$: vector of errors; $D$ is the matrix of covariance of random effects, and $R$ is the matrix of covariance of errors. The expectation and variance of $Y$ are $E(Y) = X\beta$ and $V(Y) = ZDZ^T + R = V$; therefore, $Y \sim N(X\beta, ZDZ^T + R)$. The estimation parameters are presented in [8].

$$\hat{b} = \hat{D}Z'\hat{V}^{-1}\left(y - X\hat{\beta}\right) \tag{2}$$

$$\hat{\varepsilon} = y - X\hat{\beta} - Z\hat{b} \tag{3}$$

where $\hat{\beta} = \left(X'\hat{V}^{-1}X\right)^{-} X'\hat{V}^{-1}y$ and $\hat{D}$ and $\hat{V}$ are residual maximum likelihood (REML) [9] estimates, $\hat{\beta}$ contains the best linear unbiased estimators, $\hat{b}$ contains the best linear unbiased predictors (BLUPs), and can be used to select $D$. The residual $\hat{\varepsilon}$ is used to select the $R$ matrix, and to check normality assumptions.

Mixed models allow one to select different variance-covariance structures for repeated measures experiments with or without missing data to see which covariance structure best fits the model [8]. The covariance structures used in this study are described below.

*Compound symmetry* (*CS*). The CS structure assumes that observations of the same subject have homogeneous variance and covariance. With four observations, the matrix takes the form:

$$CS = \begin{bmatrix} \sigma_1^2 + \sigma^2 & \sigma_1^2 & \sigma_1^2 & \sigma_1^2 \\ \sigma_1^2 & \sigma_1^2 + \sigma^2 & \sigma_1^2 & \sigma_1^2 \\ \sigma_1^2 & \sigma_1^2 & \sigma_1^2 + \sigma^2 & \sigma_1^2 \\ \sigma_1^2 & \sigma_1^2 & \sigma_1^2 & \sigma_1^2 + \sigma^2 \end{bmatrix} \tag{4}$$

*First order-auto regressive AR* (1). The *AR* (1) model assumes that measurements close to one another in time will show high correlations. The variances between measurements are equal, but the covariance between observations of the same subject decreases exponentially as lag increases. The correlation $\rho$ is for observations of the interval one and two, $\rho^2$ is for observations one and three, $\rho^3$ is for observations one and four, and so on. Hence, the *AR* (1) structure follows an exponential function; i.e., $corr_{AR(1)}(lag) = \rho_{AR(1)}^{lag}$.

$$AR(1) = \sigma^2 \begin{bmatrix} 1 & \rho & \rho^2 & \rho^3 \\ \rho & 1 & \rho & \rho^2 \\ \rho^2 & \rho & 1 & \rho \\ \rho^3 & \rho^2 & \rho & 1 \end{bmatrix} \tag{5}$$

*Toeplitz.* Toeplitz structure, sometimes called "banded", specifies that covariance depends on lag. The correlation function is $corr(lag) = \sigma_{Toep,|lag|}/\sigma_{Toep}^2$. The elements of the main diagonal of $R$ are $\sigma_{Toep}^2$. All elements in sub-diagonal $|k-1| = lag$ are $\sigma_{Toep,|lag|}$, where k is the row and l is the column number

$$Toep = \begin{bmatrix} \sigma^2 & \sigma_1 & \sigma_2 & \sigma_3 \\ \sigma_1 & \sigma^2 & \sigma_1 & \sigma_2 \\ \sigma_2 & \sigma_1 & \sigma^2 & \sigma_1 \\ \sigma_3 & \sigma_2 & \sigma_1 & \sigma^2 \end{bmatrix} \tag{6}$$

*Unstructured* (*UN*). In the UN model, all variances and covariances are different; that is, unstructured specifies no pattern in the covariance matrix, and it is completely general. The use of this structure requires estimation of many parameters, $k(k+1)/2$, where $k$ is the number of repeated measures.

$$UN = \begin{bmatrix} \sigma_1^2 & \sigma_{12} & \sigma_{13} & \sigma_{14} \\ \sigma_{12} & \sigma_2^2 & \sigma_{23} & \sigma_{24} \\ \sigma_{13} & \sigma_{23} & \sigma_3^2 & \sigma_{34} \\ \sigma_{14} & \sigma_{24} & \sigma_{34} & \sigma_4^2 \end{bmatrix} \tag{7}$$

### 2.3. Model Development Stages

Three stages are involved when developing a suitable foundation for model selection. The first stage involves the selection of the fixed effects (structure mean) such that we have models with the maximum information. Over-fitted models are preferred over under-fitted ones to avoid the introduction of spurious correlations [10]. The second stage consists

of selecting the covariance structure for $D$ and $R$ matrices. The third stage uses formal techniques to compare various covariance structures.

*2.4. Evaluated Models*

Univariate and multivariate analysis approaches are used for estimating and analyzing repeated measurements data. In order to estimate and predict sugarcane yield, we used five mixed model approaches that may account for most of the variation in this study, as described below.

*Model 1.* The first simple model considered is the completely randomized design (CRD), and it is described as follows:

$$y_{im} = \mu + \tau_i + e_{im} \tag{8}$$

where $y_{im}$ is the yield measured with the $i$th variety at the $m$th replication $(i = 1, 2, \cdots, 26; m = 1, 2, \cdots, v_i)$, $\mu$ is the overall mean of all observations, $\tau_i$ is the $i$th fixed effect level due to variety, and $e_{im}$ is the random error assuming $e_{im} \sim IIDN(0, \sigma^2)$.

*Model 2.* Randomized block design. Assuming that there is variation in plots within location (random factors), the statistical model is written as

$$y_{ilk} = \mu + \tau_i + plot(loc)_{k(l)} + e_{ilk} \tag{9}$$

where $y_{ilk}$ is the yield observed in the $i$th variety at $k$th plot within the $l$th location, $\mu$, $\tau_i$ and $e_{ilk}$ are as described above $(i = 1, 2, \cdots, 26; l = 1, 2, \cdots, 24; k = 1, 2, \cdots, v_{il})$, and $plot(loc)_{k(l)}$ is the random effect due to plot within location, with $plot(loc)_{k(l)} \sim IIDN\left(0, \sigma^2{}_{plot(loc)}\right)$. In matrix notation, model (9) can be written as $\mathbf{Y} = \mathbf{X\beta} + \mathbf{Zb} + \mathbf{\varepsilon}$. In this approach, the total variance-covariance matrix $V$ can be partitioned into the variance-covariance matrix for the random effects due to plot within location $b$ $[Var(b) = D]$, and the variance-covariance matrix for the random errors $e$ $[Var(e) = R]$, and so $V = ZDZ' + R$.

*Model 3.* Since the ratoon number in sugarcane variety trials is a key factor that affects stalk cane yield, it must be taken into account in the analysis. In this case, the ratoon number is taken as a class variable:

$$y_{ijkl} = \mu + \tau_i + plot(loc)_{k(l)} + \beta_j + (\tau * \beta)_{ij} + e_{ijkl} \tag{10}$$

where $y_{ijkl}$ is the yield observed in the $i$th variety, $j$th ratoon number, and $k$th plot in the $l$th location $(i = 1, 2, \cdots, 26; j = 0, 1, 2, \cdots, 8; l = 1, \cdots, 24; k = 1, 2, \cdots, v_i)$, $\mu$, $\tau_i$, $plot(loc)_{k(l)}$ and $e_{ijkl}$ are as above, and $\beta_j$ is the fixed effect due to $j$ ratoon number, $(\tau * \beta)_{ij}$ is the interaction fixed effect between variety and ratoon number. Both random errors $plot(loc)_{k(l)}$ and $e_{ijkl}$ are independent and not correlated.

*Model 4.* Analysis of covariance model in a randomized block design with the ratoon number as continuous covariable as described below:

$$5y_{ijkl} = \mu + \tau_i + plot(loc)_{k(l)} + (\beta + \delta_i)X_{ij} + e_{ijkl} \tag{11}$$

where $y_{ijkl}$ is the yield observed in the $i$th variety, $k$th plot at $l$th location with the $j$th ratoon number $(i = 1, 2, \cdots, 26; l = 1, 2, \cdots, 24; k = 1, 2, \cdots, v_i; j = 0, 1, \cdots, 8)$, $\beta$ is the intercept of the covariable for the variety $i$ having the $j$ ratoon number $(X_{ij})$, $\delta_i$ is the slope for the variety $i$, and $\mu$, $\tau_i$, $plot(loc)_{k(l)}$ and $e_{ijkl}$ as defined above. Both random errors $plot(loc)_{k(l)}$ and $e_{ijkl}$ are no correlated.

*Model 5.* Mixed model repeated measures can be seen as split plots in time [4]. Split plots in time are commonly used in agricultural studies and are often known as factorial designs. The model for this split plot in time is as follows:

$$y_{ijklm} = \mu + \tau_i + plot(loc)_{k(l)} + \tau * plot(loc)_{imk(l)} + \lambda_m + (\tau * \lambda)_{im} + \beta_i x_{ij} + e_{ijklm} \tag{12}$$

where $y_{ijklm}$ is the cane yield observed in the $i$th variety having $j$th ratoon number, $m$th crop year at $k$th plot within the $l$th location ($i = 1, 2, \cdots, 26$; $m = 1, 2, 3, 4$; $l = 1, 2, \cdots, 24$; $k = 1, 2, \cdots, v_i$; $j = 1, 2, \cdots, 8$), $\mu$, $\tau_i$ are defined as before, $\lambda_m$ is the fixed effect of crop year, $(\tau * \lambda)_{im}$ is the interaction fixed effect between variety and year, $\beta_i$ is a fixed slope coefficient of the covariable ratoon number $(x_{ij})$ on each variety, $plot(loc)_{l(k)}$ is the random error due to plot within location with $plot(loc)_{k(l)} \sim IID\left(0, \sigma^2_{plot(loc)}\right)$, $\tau * plot(loc)_{ijk(l)}$ is the whole plot error assuming $\tau * plot(loc)_{ijk(l)} \sim IIDN\left(0, \sigma^2_{\tau*plot(loc)}\right)$, and $e_{ijklm}$ is the experimental random normal error with mean zero and variance $\sigma^2$; that is, $e_{ijklm} \sim IID\left(0, \sigma^2\right)$. All term errors are noncorrelated random effects.

All data were subjected to analysis of variance using the GLIMMIX procedure of SAS 9.4. Since sugarcane varieties had different numbers of replicates in locations, the Kenward–Roger degrees of freedom adjustment [11] was used. Moreover, the effect of ratooning number on cane yield was analyzed using a polynomial orthogonal contrast. PROC GLIMMIX allows us to specify separately and jointly covariance structures that assumes heterogeneity within and/or between subjects. The CS covariance structure in PROC GLIMMIX can be specified with the RANDOM statement and the options SUBJECT and TYPE = CS. The AR (1) covariance structure is specified for each subject with the RANDOM statement and TYPE = AR (1).

The information criteria provided by PROC GLIMMIX as well as PROC MIXED are used as statistical tools to select and measure the relative fit of two or more competing models. The -2 residual log likelihood (-2RLL) [12], the Akaike information criterion (AIC) [13], the corrected Akaike information criterion (AICC) [14], the Bayesian information criterion (BIC) [15], and the mean square error (MSE) were used to compare the competing models.

The mean square error (MSE) is a measure of the difference between values predicted by a model $(\hat{y}_i)$ and the values observed from the environment that is being modeled $(y_i)$. These individual differences are residuals, and the MSE serves to aggregate them into a single measure.

$$MSE = \frac{1}{n} \sum_{i=1}^{n} (y_i - \hat{y}_i)^2 \tag{13}$$

A model with less MSE is preferred when comparing different models.

## 3. Results

### 3.1. Descriptive Statistics

The total cane yield that sugar mills received from 2012 to 2016 was 5,469,781 Mg from 24 locations with 82,259.4 ha. Table 3 shows the surface cultivated in sugarcane, which decreased in 2013–2014 by 31.69% and increased in the production cycles 2014–2015 (30.7%) and 2015 (32.62%). However, average cane yield per cycle from 2012 to 2016 decreased by close to 10%.

**Table 3.** Surface cropping, total, and average cane yield per cycle.

| Production Cycle | Surface (ha) | Yield (Mg) | Average Yield (Mg ha$^{-1}$) |
|---|---|---|---|
| 2012–2013 | 19,057.5 | 1,388,215.8 | 73.4 |
| 2013–2014 | 13,016.6 | 926,264.9 | 71.2 |
| 2014–2015 | 24,909.8 | 1,565,579.6 | 62.8 |
| 2015–2016 | 25,275.3 | 1,589,721.5 | 62.9 |
| Total | 82,259.4 | 5,469,781.9 | 67.4 |

The individual cane yields of 26 varieties were plotted against production cycle in Figure 1. There, variation in cane yield at the beginning of this study was large, an indication that cane yield decreases as variety age increases. This cane yield behavior could be due to the fact that as the age of sugarcane increases, ratoon number also increases in each cycle of production, among other factors.

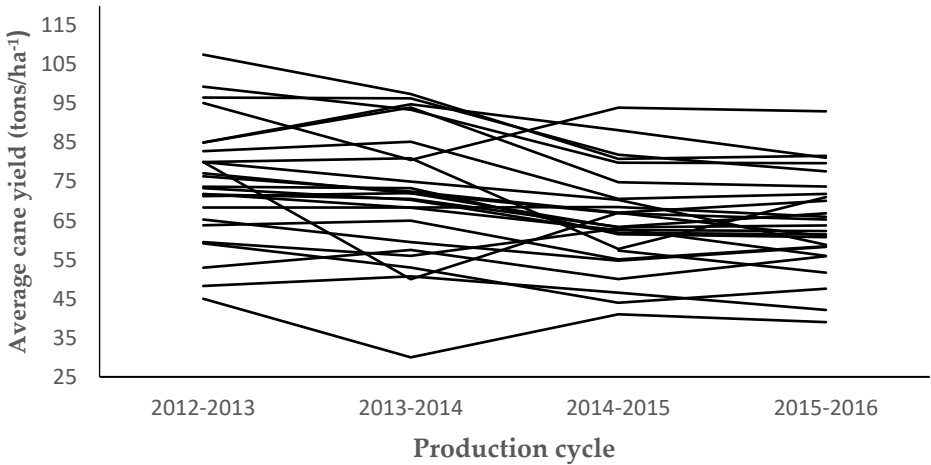

**Figure 1.** Average cane yield per production cycle of 26 sugarcane varieties.

*3.2. Model Selection*

In the definition of a mixed model, it is important that the selection of covariates define the mean structure of fixed effects and the covariates of random effects. The comparison model's fit statistics, -2 Res Log Likelihood, AIC, AICC, BIC, and CAIC, are in Table 4, for the five fitted models. The values of the fit statistics decreased from models 1, 2, 5, 4 and 3 as more covariates and random effects in the linear predictor were appropriately added, but in model 5, these fit statistics increased little. We observed that when the plot within location effect and the covariable ratoon number were included in models 2–5, these factors had a significant impact on the fit statistics compared to the CRD model. This comparison is interesting because the dynamics and characteristics of varieties on cane yield are different from those of the ratoon crop. Therefore, according to all fit statistics, the best model was model 3. Data suggest that there is significant spatial variability in each plot location as well as in the ratooning number of sugarcane varieties (Table 4), and, therefore, when the aforementioned factors were included in the linear predictor, the estimates of cane yield. improved.

**Table 4.** Fit statistics (smaller is better) for different model approaches.

| | Model Approach | | | | |
|---|---|---|---|---|---|
| **Fit Statistics** | **(1)** | **(2)** | **(3)** | **(4)** | **(5)** |
| -2 Res Log Likelihood | 387,416.1 | 377,760.3 | 376,031.9 | 376,484.9 | 377,713.8 |
| AIC | 387,470.1 | 377,764.3 | 376,035.9 | 376,490.9 | 377,719.8 |
| AICC | 387,470.1 | 377,764.3 | 376,035.9 | 376,490.9 | 377,719.8 |
| BIC | 387,705.1 | 377,773.7 | 376,052.6 | 376,505.0 | 377,733.9 |
| CAIC | 387,732.1 | 377,775.7 | 376,054.6 | 376,508.0 | 377,736.9 |

*3.3. Mean Square Error (MSE) between Models*

The MSEs were plotted in order to compare between our different approaches. Parallel to fit statistics, the MSE decreased as we redefined the linear predictor to obtain valid inferences for the fixed effect parameters (Figure 2). This result indicates that when fitting mixed models, an appropriate mean structure and random effects need to be specified, in order to take into account most of the unexplained variation that is due to either within locations–varieties or between locations–varieties associated with levels of ratooning, as in models 4 and 5.

A key strength of these approaches is the ability to control factors and settings that minimize the effects of the uncontrollable factors such as plot, location, ratooning number, and time. For example, we reduced the MSE from model approach 1 to model approach 2 by 23.55% (348.24 to 266.28); approach 2 to 3 was 62.35% (266.28 to 100.25), but when we included the production cycle (year) in the model, the MSE increased in models 4 to 5 from 110.88 to 238.22. The increase in the MSE may be due to the fact that the analyzed varieties have different production years (ratooning number) and a different number of repetitions in locations that could explain why approach 5 gave larger MSE compared with approaches 3 and 4. Under these approaches, models 3 and 4 are quite similar when we set the ratoon number as classification or continuous variable in the linear predictor.

The estimates of mean yields (Mg ha$^{-1}$) and standard errors of means for each variety and model approach as well as the global average of mean yields and average of standard errors of means (bold letters), are shown in Table 5. Models 4 and 5 gave the highest average standard error with values of 3.99 and 4.57, respectively, while model approach 3 gave the lowest average standard error. According to the fit statistics and MSE, the best model to estimate cane yield of sugarcane varieties was model 3. Table 5 was descendent ordered by mean estimation of model 3, and the three varieties with highest mean yield were ITV 92-373, ATEMEX 96-40, and L 77-50, but the variety ITV 92-373 had a large standard error, which rendered the varieties ATEMEX 96-40 and L 77-50 more preferable.

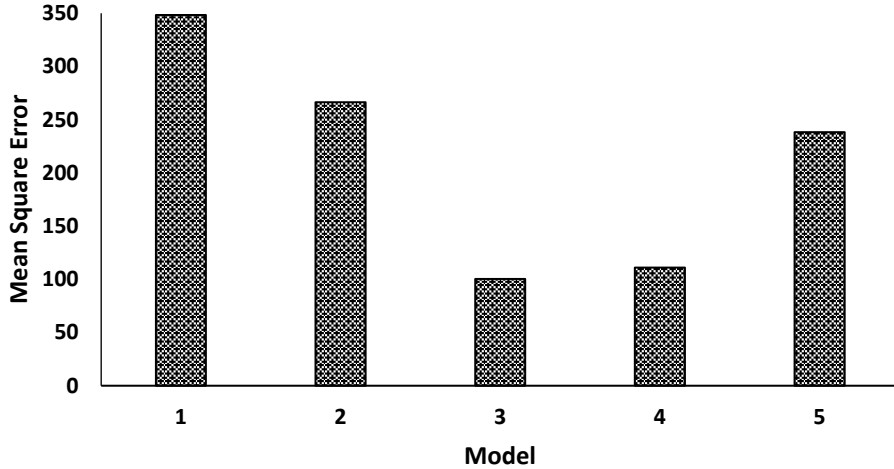

**Figure 2.** Mean square error of model approaches.

### 3.4. Ratoon Crop

Among different agricultural practices that may affect cane yield, the management of the ratoon crop in sugarcane is a key factor. Growing ratoon crops cost less than planting cane and, therefore, achieving high-yield ratoon crop is a valuable objective; this crop is more economical than seed material, reducing operational costs by 25–30%. In addition, ratoon productivity may improve cane stalk yield with proper management involving timely agricultural operations, proper nutrition management, integrated pest management and maintenance of an adequate plant population. In our study, the quadratic effect of the ratoon number on the cane yield was estimated ($P = 0.0001$). Figure 3 shows that stalk cane yield tends to decrease by around 10% as the number of ratoon crops increases, since most sugarcane varieties have more than 5 ratoon crops.

**Table 5.** Mean sugarcane yield (Mg/ha) and standard error of mean for each model and sugarcane variety.

| | Model Approach | | | | | | | | | |
|---|---|---|---|---|---|---|---|---|---|---|
| | (1) | | (2) | | (3) | | (4) | | (5) | |
| Variety | Estimate | Standard Error | Estimate | Standard Error | Estimate | Standard Error | Estimate | Standard Error | Estimate | Standard Error |
| ITV 92-373 | 105.33 | 10.78 | 98.47 | 9.51 | 84.99 | 8.51 | 96.61 | 9.33 | 95.37 | 9.95 |
| ATEMEX 96-40 | 86.66 | 1.51 | 82.03 | 1.42 | 78.41 | 1.28 | 79.26 | 3.05 | 72.92 | 3.50 |
| L 77-50 | 83.1 | 0.85 | 82.3 | 0.87 | 77.03 | 0.74 | 87.30 | 2.37 | 89.36 | 2.60 |
| M.Y. 5514 | 87.34 | 2.23 | 88.03 | 2.03 | 76.15 | 1.86 | 76.78 | 4.27 | 90.96 | 4.98 |
| BCO. DE VARS. | 90.94 | 2.61 | 78.28 | 2.55 | 75.32 | 2.66 | 67.09 | 3.87 | 73.17 | 6.02 |
| ITV 92-1424 | 75.64 | 0.52 | 77.53 | 0.63 | 72.71 | 0.46 | 71.93 | 0.82 | 73.32 | 1.10 |
| LGMex 92-156 | 75.5 | 3.20 | 77.77 | 2.97 | 70.95 | 2.38 | 64.38 | 3.89 | 60.73 | 4.40 |
| CP 72-2086 | 71.65 | 0.29 | 71.07 | 0.49 | 70.73 | 0.30 | 70.82 | 0.30 | 69.71 | 0.53 |
| MEX 73-523 | 61.6 | 8.35 | 56.16 | 7.59 | 69.90 | 6.72 | 69.77 | 7.51 | 56.98 | 8.03 |
| MEX 79-431 | 69.94 | 0.25 | 69.00 | 0.47 | 69.49 | 0.26 | 69.18 | 0.25 | 68.54 | 0.50 |
| MEZCLA PREC | 67.09 | 0.24 | 68.32 | 0.46 | 68.34 | 0.25 | 67.35 | 0.29 | 67.28 | 0.52 |
| MEX 56-476 (P-01) | 69.55 | 5.63 | 70.92 | 5.02 | 67.99 | 4.82 | 65.36 | 5.00 | 71.15 | 5.32 |
| CP 72-1210 | 66.64 | 1.25 | 70.38 | 1.19 | 67.98 | 1.08 | 68.02 | 1.09 | 70.13 | 1.39 |
| RD 75-11 | 67.02 | 0.69 | 73.60 | 0.77 | 67.88 | 0.62 | 67.13 | 0.74 | 71.87 | 1.05 |
| Mezcla Media | 65.23 | 0.26 | 66.52 | 0.47 | 66.98 | 0.27 | 66.93 | 0.30 | 67.08 | 0.54 |
| PR 1013 | 66.43 | 4.99 | 68.92 | 4.48 | 66.71 | 4.39 | 64.20 | 4.53 | 69.60 | 4.96 |
| MEX 69-290 | 65.95 | 0.09 | 67.42 | 0.41 | 66.64 | 0.14 | 66.33 | 0.13 | 66.75 | 0.41 |
| MEX 56-18 | 57.44 | 6.22 | 61.71 | 5.53 | 64.87 | 4.65 | 62.95 | 7.28 | 72.15 | 8.99 |
| MEX 57-473 | 61.08 | 2.95 | 58.72 | 2.68 | 64.20 | 2.71 | 66.37 | 4.95 | 55.37 | 4.90 |
| CP 70-1133 | 68.11 | 6.22 | 61.15 | 5.53 | 63.51 | 5.08 | 58.95 | 9.17 | 50.24 | 9.51 |
| MEX 68-P-23 | 59.39 | 0.64 | 66.89 | 0.72 | 63.18 | 0.58 | 63.12 | 0.58 | 65.52 | 0.82 |
| CP 44-101 | 52.94 | 0.93 | 64.14 | 0.96 | 60.94 | 0.79 | 60.27 | 0.85 | 63.06 | 1.23 |
| SP 70-1284 | 59.5 | 13.2 | 59.45 | 11.98 | 56.94 | 9.88 | 22.15 | 16.11 | 43.64 | 17.8 |
| Co 997 | 49.84 | 3.73 | 60.54 | 3.54 | 55.61 | 3.48 | 54.68 | 3.77 | 58.70 | 4.11 |
| P.O.J 2878 | 47.18 | 1.34 | 65.41 | 1.51 | 50.58 | 1.27 | 52.63 | 2.14 | 60.86 | 3.22 |
| B 43-62 | 38.75 | 9.33 | 48.49 | 8.33 | 40.22 | 9.23 | 39.96 | 11.21 | 40.14 | 12.45 |
| Average | 68.07 | 3.40 | 69.74 | 3.16 | 66.86 | 2.86 | 65.37 | 3.99 | 67.10 | 4.57 |

Estimate: Estimated mean (Mg ha$^{-1}$). Standard Error: Standard error of the mean.

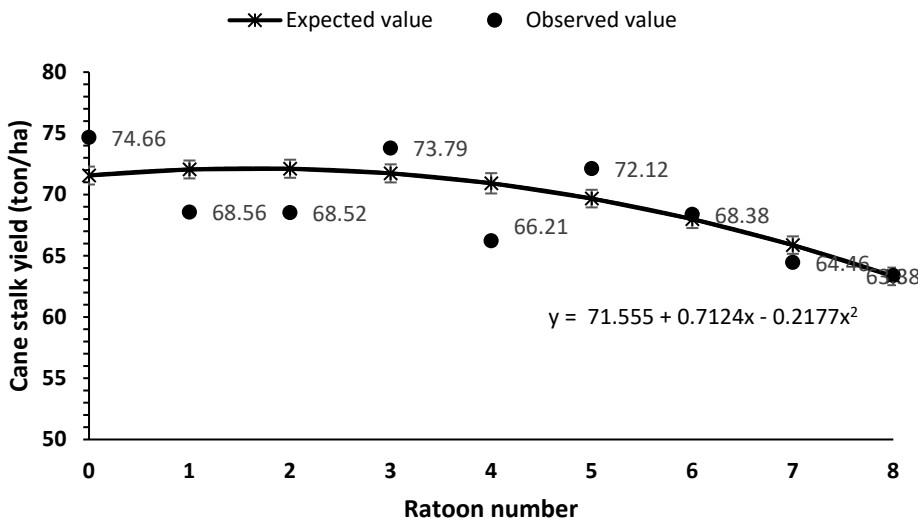

**Figure 3.** Effect of the number of ratoon crops on cane stalk yield.

## 4. Discussion

Researchers often ask how to build an appropriate mixed effects model. Our study described a systematic step-by-step strategy including different mixed model approaches and practical advice for achieving good estimates when estimating stalk cane yield. This is possible due to statistical programs such as PROC MIXED and PROC GLIMMIX, which allow one to define different mean structures to be included in the model.

The correct translation from the data to the statistical model requires stating and discussing the implications of the limitations of the data relative to the aims of the study. Therefore, the selection of a model with the maximum information depends on statistical and scientific considerations in the specific field. For example, estimating and predicting yield in sugarcane variety trials would need to include the following in the model: genotypes (varieties), number of ratooning, location, plot, and time, among others. From a statistical point of view, a large-good model could be chosen to avoid omitting a significant factor (or covariate) and to maximize the estimation and prediction power (avoiding type II error). However, a simple model could be chosen to avoid including a nonsignificant factor (or covariate) and hence to maximize the reliability and parsimony of the model.

Researchers want a simple model to explain most of the variation present in the data. When assessing the goodness of fit of a model, they usually compare different candidate models to see whether one of them offers the best fit. Three scenarios in a mixed model may appear: (1) mean comparison of the models with the same covariance structure—nested models are the most common; (2) compared covariance models with the same linear predictor; and (3) compared mixed models with different linear predictors and different covariance structures.

A likelihood ratio test (LRT) can be used to compare the reduced model against the full model. However, it can also use the REML method by maximizing the likelihood function of a set of error contrasts to account for the loss in the degrees of freedom involved in the estimation of fixed effects, because changing the fixed effects in the REML estimation leads to a different design matrix $X$. Thus, the difference in $-2$ restricted maximum log likelihood values between nested effect models does not produce a valid LRT [4]. Now, when comparing nested models with different covariance structures but the same fixed effects (mean), the REML functions from the two models are comparable since they have the same mean structure [12]. However, sometimes neither the LRT through maximum likelihood (ML) method nor REML method can be used [16].

For comparing and selecting the best candidate model, many information criteria were proposed as alternatives to LRT, specifically when the models are not nested. The main idea with the information criteria is to compare models with different covariance

structures and the same fixed effects and with their maximized log likelihood values, but penalizing the use of too many parameters in the model [4,10,17]. The criteria are ordered in increasing preference of parsimony, and the model with the smaller information criteria values provides the better fit. Finally, once the candidate model is chosen, the covariance structures for models are determined using the Kenward–Roger adjustment degrees of freedom [11].

The selection of a mean structure that gives better estimates to fit statistics plays an important role in defining the fixed and random effects in the mixed model. When specifying random effects, two aspects should be highlighted: the variation between plots, locations, and plots within locations; and the type of covariance structure between measurements (cane yield between crop years). In our study, different mean structures and random effects in the mixed model gave different estimates as well as standard errors of these estimates. These estimates can vary widely when sources of variation are not appropriately specified in the model. According to the data from our research and model 3 for the estimations, a recommendation for farmers would be to remove the plots with varieties that yield less than $60\,\mathrm{Mg\,ha^{-1}}$ and use varieties with a yield greater than $75\,\mathrm{Mg\,ha^{-1}}$. Moreover, Viator et al. [18] showed that harvest time and ratoon crop affects ratooning ability and consequently cane yield. Likewise, Matsuoka and Stolf [19] said that cane yield declines with increasing numbers of ratoons and the replacement of a new plantation is necessary, but with good agronomic practices the cane yield might be maintained; for example, Nuss [20] and Kingston [21]) reported 20 to 25 successive harvests. Thus, the ratooning crop is an important factor that affects sugarcane production; thus, paying attention to its management and engaging in good agricultural practices may bring great economic benefits because of the high cost for establishing a new sugarcane plantation.

## 5. Conclusions

The ability of mixed models to account for most of the variation due to sugarcane varieties, plots, crop years, ratoon number and locations was demonstrated. The estimates of cane yield when including the covariable "ratoon number" either as class variable (model 3) or as continuous variable (model 4) produced the lowest standard errors for the mean cane yield.

The relevance of having estimates with a lower degree of uncertainty will allow making better decisions regarding which varieties that should continue to be planted on the field as well as how much sugar will be obtained in a certain crop year to make marketing decisions.

When estimating cane yield in sugarcane trials, the ratoon number should be included in the mixed model as a continuous or classification variable to achieve better estimates, since ratooning is an important cultivation practice in sugarcane production worldwide, with underground buds on the remaining stem acting as a source for the establishment of a subsequent ratoon crop. In addition, the use of the appropriate mixed model may facilitate better judgments when recommending which sugarcane varieties should be on the fields for production. Moreover, these results confirm the relevance of using an appropriate mixed model in estimating or predicting cane yield, or even more, when selecting sugarcane genotypes for local adaptations with the objective of enhancing the mean productivity of the whole cane industry.

**Author Contributions:** Conceptualization, S.L.H.-V. and J.S.-R.; methodology, J.S.-R.; software, J.V.H.-C. and J.S.-R.; formal analysis, J.V.H.-C.; data collection and data curation, S.L.H.-V.; writing—original draft preparation, J.S.-R.; writing—review and editing, J.M.R.-P. All authors have read and agreed to the published version of the manuscript.

**Funding:** This research received no external funding.

**Institutional Review Board Statement:** Not applicable.

**Informed Consent Statement:** Informed consent was obtained from all subjects involved in the study.

**Data Availability Statement:** Not applicable.

**Conflicts of Interest:** The authors declare no conflict of interest.

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
