# Peer review of "Selection and Fitting of Mixed Models in Sugarcane Yield Trials"

_agriculture, doi:10.3390/agriculture12030416_

Round 1

Reviewer 1 Report

Introduction

  1. From line 60, please perform an analysis of research with the use of

mixed model methodology: cite authors' names, compare results and complete literature reports.

  1. Necessarily, as new part of the text  (last paragraph of introduction) - formulate the aim of the the work, please.

Materials and methods

  1. Too poor information on the tested material (point 2.1., Lines 71 to 73) - sugar cane variety, basic properties and others - see other similar publications.

Generally, characteristics of the research material and information about its origin and cultivation are missing.

  1. Display the equations used in the calculations from the paper, give the origin of these equations (references, explanations), explain all symbols. Highlight the next models with the bold.

Results and Discussion

  1. Standardize and bold axis descriptions in all diagrams.
  2. Take care of the graphic design of your article - for example, the form of diagram 1 is unacceptable - correct it, please.
  3. Check that you have referred to all diagrams and tables in the text.
  4. Complete the description to Table 5 (is too laconic).
  5. In point 3.3 refer to data contained in the literature, it is important in a scientific publication.

Conclusion

Expand the last sentence of conclusions. Highlight the practical aspect and the possibility of using the results of your research.

Author Response

the response to the reviewer 1 is the file named Respose to reviewer 1.

Reviewer 2 Report

The current field study is interesting and falling within the scope of the journal. The manuscript presented results of the field study “Selection and fitting of mixed models in sugarcane yield trials”. There is well consistency between the title of the article and the results, and their interpretation. The write up of the manuscript is also satisfactory. The manuscript dataset certainly contain constructive information for scientific community.

The following points may be addressed by the Authors to enhance the worth of the paper.

Abstract

The abstract part is not well described. The author should also describe the results in the abstract part to improve the worth of the manuscript. The abstract part is also too short, not describing the manuscript in a proper way.

Introduction

The introduction part is also written satisfactorily. The authors should also provide information about the drawbacks of the sugarcane ratoon crop, not only the benefits of rationing. The authors have also not described models and their working in deatils. Please give a paragraph about models functioning and their benefits in agriculture.

Materials and Methods

The materials and methods are well presented.

Results and Discussion

The presented results and discussion is also satisfactory described based on objectives of the present research work but further, it can also be improved.

Conclusion

The conclusion part is also well written based on the objectives of the field study.

Specific Comments

Line 41-47- Reference source not found.

Line 30-33- No reference source is cited. The authors should cite the proper reference.

Line 53-59- Reference source not found.

Line 64-66- Cite the proper reference source.

Line 380- Author has cited too old reference. Please cite the recent reference.

Line 389,390, 397- Reference is too old.

Line 402, 403, 407- Cite the latest reference.

Author Response

Response to the reviewer 2 were done in text and other in the following specific comments:

Most of reviewer comments were included in the main text.

Most comments were added in the text.

Specific comments

Line 389,390, 397- Reference is too old.

Response: Although these references are old all them are classic references that contain an specific background and theory in which most stat methods have been developed.

Line 402, 403, 407- Cite the latest reference.

Response: These reference are a  classic papers that I could not find a recent reference.

Round 2

Reviewer 2 Report

The majority of the comments have been added.